# Investigating the Vital Role of the Identified Abietic Acid from *Helianthus annuus* L. Calathide Extract against Hyperuricemia via Human Embryonic Kidney 293T Cell Model

**DOI:** 10.3390/molecules28135141

**Published:** 2023-06-30

**Authors:** Huining Dai, Xiao Xu, Wannan Li, Xueqi Fu, Weiwei Han, Guodong Li

**Affiliations:** 1Engineering Research Center of Bioreactor and Pharmaceutical Development of Ministry of Education, College of Life Science, Jilin Agricultural University, Changchun 130118, China; daihn18@mails.jlu.edu.cn; 2Edmond H. Fischer Signal Transduction Laboratory, School of Life Sciences, Jilin University, Changchun 130012, China; x1058434572z@163.com; 3Key Laboratory for Molecular Enzymology and Engineering of Ministry of Education, Jilin University, Changchun 130012, China; fxq@jlu.edu.cn (X.F.); weiweihan@jlu.edu.cn (W.H.); 4Department of General Surgery, The Second Hospital of Jilin University, Changchun 130041, China; liguodong@jlu.edu.cn

**Keywords:** sunflower calathide, abietic acid, human embryonic kidney 293T cells, transcriptomics, xanthine oxidase inhibitor, purine metabolism

## Abstract

To explore the anti-hyperuricemia components in sunflower (*Helianthus annuus* L.) calathide extract (SCE), we identified abietic acid (AA) via liquid chromatography–mass spectrometry and found an excellent inhibitor of xanthine oxidase (IC_50_ = 10.60 µM, Ki = 193.65 nM) without cytotoxicity. Based on the transcriptomics analysis of the human embryonic kidney 293T cell model established using 1 mM uric acid, we evaluated that AA showed opposite modulation of purine metabolism to the UA group and markedly suppressed the intensity of purine nucleoside phosphorylase, ribose phosphate pyrophosphokinase 2, and ribose 5-phosphate isomerase A. Molecular docking also reveals the inhibition of purine nucleoside phosphorylase and ribose phosphate pyrophosphokinase 1. The SCE exhibits similar regulation of these genes, so we conclude that AA was a promising component in SCE against hyperuricemia. This present study provided a novel cell model for screening anti-hyperuricemia natural drugs in vitro and illustrated that AA, a natural diterpenoid, is a potential inhibitor of purine biosynthesis or metabolism.

## 1. Introduction

Human metabolic disorder hyperuricemia (HUA) has been a global public health issue with a young and fast-expanding trend. As a result of exogenous and endogenous purine metabolism, 80 percent of uric acid (2,6,8-trihydroxy purine, UA) is generated via purine nucleotide biosynthesis using de novo and salvage pathways [1]. It is generally accepted that purine catabolism disorder and UA excretion dysfunction cause hyperuricemia [2], and approximately 60–70 percent of UA is excreted or filtered through the renal tubule and glomerular. Although clinical treatments such as urate reabsorption transporter 1 (URAT1) inhibitor benzbromarone are effective, the urate excretion promoter probenecid and the xanthine oxidase (XO) inhibitors allopurinol and febuxostat are widely recognized, as demonstrated in chronic kidney patients with HUA who developed renal injury all year round [3,4]. Therefore, natural alternative drugs against hyperuricemia that have few adverse effects will urgently be investigated.

Sunflower (*Helianthus annuus* L.) calathide, which possessed medicinal and edible properties, could reduce UA levels and inhibit XO activity in hyperuricemia mice and ameliorate inflammation in gouty arthritis rats [5,6]. Several bioactive components of sunflower calathide water extract (SCE) have been identified, and they had been confirmed to prevent urate-induced renal injury [7]. Subsequently, we have identified scopoletin and chlorogenic acid as anti-hyperuricemia components of sunflower calathide [8]. However, many ingredients are expected to be explored, for example, the abietic acid (AA). It is well known for its anti-inflammatory [9,10], anti-cancer [11,12], and antioxidant properties [13]. Moreover, it was found to inhibit soybean 5-lipoxygenase [14] and cure acute lung injury [15] and atopic dermatitis [16]. Given the above abundant bioactivities, we aim to investigate whether AA has the potential for anti-hyperuricemia and compared the mechanism with SCE.

Although SCE’s anti-hyperuricemia effect has been determined, due to the presence of uricase, mouse models failed to simulate the underlying mechanism in humans well [17]. Given that the uricase-deficient mouse may disrupt some biological processes involving UA biosynthesis and metabolism, developing a human-derived cells model for screening in vitro may be beneficial. The kidney is a targeted key organ for urate transport and excretion [18]. Therefore, we supposed that human embryonic kidney (HEK) 293T cells, a derivation of 293T cell line that expresses urate transporters, including URAT1 and glucose transporter 9 (GLUT9), in a stable manner [19,20], can be evaluated to establish UA-stimulated models in vitro [21].

In this present study, the investigation of AA on UA-induced 293T cells was performed using RNA sequencing, and its inhibition of XO was evaluated. To compare AA with SCE, visual analysis was carried out via weighted gene co-expression network analysis (WGCNA) [22,23], Kyoto Encyclopedia of Genes and Genomes (KEGG) pathway [24], and gene ontology (GO). The differentially expressed genes (DEGs) may provide a better view of UA biosynthesis in the human kidney, as well as speculate how SCE and abietic acid fight against hyperuricemia.

## 2. Results

### 2.1. Identification of Abietic Acid

Under the UPLC-Q-Orbitrap HRMS, the standard and sample yielded retention times and fragments. As can be seen, the retention time of AA in the sample (Figure 1b) (Rt_Sample_ = 19.45 min) was consistent with that of the standard (Figure 1a) (Rt_STD_ = 19.36 min), and mass spectrometry yielded a [M + H]^+^ ion peak at mass to charge ratio (*m/z*) of 303.23, which agrees with standard and theoretical value, revealing that the compound was exactly abietic acid. The relative abundance of abietic acid reached 0.042 (MA_sample_/MA_standard_) in the manual integration of peak area (MA), as displayed in Figure 1. The content was calculated and converted to 0.0042 µg/mg based on sunflower calathide powder.

### 2.2. Inhibition of XO Activity

The absorbance, which indicates the formation of UA, was measured at OD290 nm after the enzymatic reactions. As shown in Figure 2a, the relative absorbance of various concentrations declined in a dose-dependent manner. And the curve of inhibition rate (Figure 2b) illustrated an efficient inhibition of XO, as well as the IC_50_, which was calculated to be 10.60 µM.

### 2.3. Docking Analysis

The docking was performed with AutoDock Tools 1.5.6, which promptly generated docking input files. The negative value for binding energy indicates a stable structure that released energy. The relatively lowest binding energy (−9.16 kcal/mol) and inhibition constant (193.65 nM) for the docking state of ten runs were shown in Table 1. Furthermore, the best state of AA-XO complex docking was selected for visual analysis.

For further analysis of interactions, a spherical view of the active pocket of the AA-XO complex was generated via PyMOL (Figure 3). Subsequently, the residues surrounding AA within 4.0 Å were screened and displayed as sticks. Critical active site residues Glu1261, Glu802, Phe798, Phe911, Phe914, and Arg912 were discovered close to the molybdenum center, which was the catalytic base of xanthine-xanthine oxidase [25]. It formed four hydrophobic interactions, Ala910, Ala1078, Phe914, and Met1038 (Pi-Alkyl), as well as an attractive charge interaction towards Arg912. Moreover, several van der Waals contributed to the binding stability. The docking results were consistent with the experimental data, indicating that AA could efficiently inhibit XO activity.

### 2.4. Cell Proliferation

We evaluated the cell proliferation using CCK-8 assay, which assessed the cell proliferation/cytotoxicity based on the amounts of formazan product. Six concentrations (264, 132, 66, 33, 16.5, and 8.25 µM) exhibited no decline in absorbance, implying no cytotoxicity (Figure 4a). There were significant increases at concentrations of 264 and 132 µM compared to the control group.

### 2.5. Pearson’s Correlation Coefficient and Soft Threshold

Pearson’s correlation coefficient was used to validate the reliability of the sequencing data. As illustrated in Figure 4b, the correlation coefficients of intragroup samples were above 0.92, implying that the repeatability of this study was satisfactory. Moreover, correlation coefficients between intergroup samples (R^2^ > 0.8) were relevant, indicating that the dataset was highly reliable and reasonable.

A proper soft threshold plays an important role in the construction of the WGCNA network to obtain results that best fit the scale-free network distribution. As a result, the weighted correlation coefficients between genes were specified at 0.8, and the corresponding soft threshold was defined at 16 (Figure 4c), whereas the mean connectivity was near 0 (Figure 4d).

### 2.6. Weighted Gene Co-Expression Network Analysis (WGCNA)

The cluster dendrogram displayed twelve modules after merging modules with a dissimilarity threshold of 0.25, which represented green-yellow, pink, purple, turquoise, red, yellow, black, blue, brown, green, magenta, and grey (Figure 5a). Meanwhile, the eigengenes were classified into 11 modules (excluding the grey module) based on similar expression patterns. The modules were divided into four main clusters (Figure 5b): (1) black, blue, and brown modules, (2) green-yellow, pink, purple, and turquoise modules, (3) red and yellow modules, and (4) green and magenta modules. All of the above had similar expression patterns in the clusters. As observed in Figure 5c, the eigengene of cluster (2) had positive associations with the AA groups, and the turquoise module showed obvious changes after AA pretreatment (*P_(Average)_* = 0.107). Interestingly, cluster (3) showed a converse tendency between the AA and UA groups (positive correlations). The positive correlation between the SCE and AA treatments deserved concern because of the blue module’s observation, which was opposite to the control and UA groups.

### 2.7. Bioinformatics Analysis

Venn diagram showed 59 overlapping KEGG pathways among the green-yellow, pink, purple, and turquoise modules for further systematic analysis (Figure 6a). A total of 37 DEGs of purine metabolism, one of the overlapping pathways (Figure 6b), were statistically analyzed, and most of them were significantly upregulated in the AA treatment according to the clustering heatmap (Figure 6c). Accordingly, DEGs identified from the pentose phosphate pathway in the pink and turquoise modules (Figure 6d,e) exhibited a significant up- or down-regulation in the AA group, whereas they were opposite to those of the UA group. In addition, there were no obvious differences between the control and UA groups (Figure 6c).

Twenty genes involving purine metabolism, as well as five genes in the pentose phosphate pathway, were carried out from the red and yellow modules (Figure 7a). The majority of DEGs appeared to be activated in the UA group, whereas they were mostly inhibited in the SCE and AA treatment (Figure 7b,c). Furthermore, these downregulated genes are mainly responsible for the biological process, including the nucleotide metabolic process according to the GO enrichment (Figure 7d). As for the blue module, eighteen genes related to purine metabolism were carried out, especially the purine nucleoside phosphorylase (PNPase), which is a crucial enzyme for purine biosynthesis and significantly downregulated after AA treatment (Figure 7e).

Moreover, we also focused on Ribose-phosphate pyrophosphokinase 1 (PRPS1), Ribose-phosphate pyrophosphokinase 2 (PRPS2), and Ribose 5-phosphate isomerase A (RPIA), which were identified from other modules and participated in the pentose phosphate pathway. The PRPS1 and PNPase were highly transcribed in the HEK293T cells according to the FPKM value (Figure 8a). When compared to the control group, PRPS2, PNPase, and RPIA showed the most significant decline in AA treatment, whereas there was a little decrease in the UA group (Figure 8a). They co-regulate small molecule metabolic processes according to the enriched GO annotation. The RPIA, in particular, is responsible for 15 biofunctions, making it a key for purine metabolism (Figure 8b). According to docking results, AA could interact with PRPS1 and PNPase, which bound with Arg-96 and Glu-205 in a distance of 2.1 Å and 1.9 Å, respectively (Figure 8c,d).

## 3. Materials and Methods

### 3.1. Preparation of Sunflower Calathide Hydrolyzed Extract (SCE)

The botanical sample of sunflower calathide was collected from Baicheng City, Jilin Province (123°12′45″ E, 44°52′23″ N) and authenticated by Professor Shuwen Guan (School of Life Sciences, Jilin University). The sunflower calathide powder was extracted via cellulase and CaCl_2_, followed by filtration. The detailed procedure refers to our previous study [8]. Extract was stored at −80 °C and re-suspended according to experiments.

### 3.2. Characterization of Extract via UPLC-Q-Orbitrap HRMS

Ultra-high-performance liquid chromatography coupled with Q-Orbitrap (UPLC-Q-Orbitrap HRMS, Thermo Fisher Scientific, Waltham, MA, USA) was previously applied to speculate the components of SCE [7]. Standard identification of abietic acid (HPLC ≥ 98%, Solarbio, Beijing, China) is essential for the rigor of this present study. The sample (200 mg) was dissolved in methanol solution (methanol: water, 8:2, *V*/*V*; 1 mL) before centrifugation at 20,000× *g* for 10 min. Subsequently, impurities were removed using a 0.22-µm PES membrane. Abietic acid was diluted with methanol and sonicated for 5 min to obtain a standard stock solution with a concentration of 20 µg/mL. The mass spectrometry was performed via ESI with positive and negative ion switching scanning. For chromatography, it was accomplished according to gradients of aqueous and organic phases. More details can be obtained in our previous study [8]. The quantification of AA was determined using peak area calculation.

### 3.3. Spectrophotometer-Based Determination of Enzyme Activity

As far as we know, the spectrophotometric method has been widely used to evaluate enzymatic reaction products; thus, we measured UA products via OD290 nm (optical density at 290 nm) [26]. The abietic acid was dissolved (Solarbio, BJS, CHN) in ethanol and diluted it into six gradients of final concentration (660, 330, 165, 82.5, 41.25, and 20.63 µM, respectively). The enzyme reaction system included PBS (1/15 M), substrate (0.5 mM of xanthine; Aladdin, CHN), and enzyme (0.05 U/mL of XO; Roche, Basel, Switzerland), and inhibitor (AA) was maintained at 25 °C. After that, the relative generation of UA was eventually calculated using the following formula:(1)Inhibition=1 − (A1 − A2/A3−A4)×100%

A1: OD290 nm of the solution containing inhibitor, xanthine, and XO.

A2: OD290 nm of the solution containing inhibitor and xanthine.

A3: OD290 nm of the solution containing xanthine and XO.

A4: OD290 nm of the solution only containing xanthine.

### 3.4. Molecular Docking

Briefly, the crystal structures of XO (PDB ID: 3NVW), PNP (PDB ID: 1RCT), and PRPS1 (PDB ID: 2H06) were obtained from the Protein Data Bank, and the 3D structure of AA was retrieved from ChemSpider (ChemSpider ID: 10127). AutoDock Tools (ADT) 1.5.6 was performed to eliminate water molecules and heteroatoms, as well as add hydrogens for semi-flexible docking. The grid box was set at 48 × 48 × 50 points to 0.375 Å spacing for calculation, and ten docking runs were performed by AutoDock 4 along with AutoGrid 4.2.6 software (Numeca International, Belgium, 2009). Additionally, the docking of PNP and PRPS1 was performed using AutoDock Vina. The docking results were visualized using PyMOL or Discovery Studio.

### 3.5. Cell Culture

The HEK293T cells were obtained from the cell bank of the Chinese Academy of Sciences (Shanghai, CHN), followed by cultured in DMEM high glucose medium (HyClone, USA) with 10% fetal bovine serum (FBS; ABW, URY) and 1% penicillin/streptomycin (Solarbio, CHN). Abietic acid (Solarbio, CHN), UA, and SCE were dissolved with DMEM and diluted. In brief, the cells were divided into four groups (Control, UA, SCE, and AA) and incubated at 100 mm Petri dishes to a density of 5 × 10^6^ cells/plate before being pretreated with SCE (300 µg/mL) and AA (132 µM) for three hours. Subsequently, 1 mM UA (Sigma-Aldrich, St. Louis, MO, USA) was performed for 24 h to establish a high uric acid model, except for the control group. All of the above were final concentrations. All the cells were maintained at 37 °C with a humidified atmosphere containing 5% CO_2_. Afterward, cells were collected for total RNA extraction (Sangon, Shanghai, China). The cytotoxicity/cell proliferation was measured at 450 nm using Cell Counting Kit-8 (CCK-8) assay kits (Sangon, CHN) after a 24 h incubation with six concentrations of AA in a 96-well plate.

### 3.6. Construction of cDNA Library and Transcriptome Sequencing

RNA extraction was prepared based on the instruction manual of TRIzol reagent (Sangon Biotech, China) and sent to Novogene Technology Co., Ltd. (Beijing, China) for cDNA library construction and RNA sequencing.

### 3.7. Datasets

The reference genome of *Homo sapiens* and gene annotation files were downloaded from the National Center for Biotechnology Information (NCBI). The reference genome index was then constructed and contrasted to the paired-end clean reads using Hisat2 v2.0.5. StringTie (v1.3.3b) [27], which was applied to assemble the mapped reads and predict novel transcripts, as well as using featureCounts v1.5.0—p 3 to evaluate gene expression levels. The DESeq2 R package (1.20.0) provided statistical routines for differential expression analysis. Genes with adjusted *p*-value < 0.05 and |log2foldchange| > 0 were considered differentially expressed genes.

### 3.8. Bioinformatic Analysis

Pearson’s correlation coefficient was used to examine the reliability of the experiment as well as the rationality of sampling via the R (Version 3.0.3) ggplot2 package. The DEGs were statistically enriched in KEGG pathways using the R (Version 3.0.3) clusterProfiler package, followed by GO enrichment. The DEGs were subjected to the Weighted Gene Co-Expression Network Analysis (WGCNA) R package [28] for complicated data and various weighted association analyses. We also used the R (Version 3.0.3) ggplot2/pheatmap package for the DEGs cluster and heatmap on the NovoMagic platform.

### 3.9. Statistical Analysis

The experimental data were analyzed via one-way ANOVA analysis using GraphPad Prism Version 8.2.1 for Mac (GraphPad Software 8.2.1, La Jolla, CA, USA) and no less than three independent experiments. Bar graphs are represented as mean ± standard deviation (SD). *p* < 0.05 was considered statistically significant. Bioinformatics analysis diagrams were performed on the NovoMagic platform, and docking results were analyzed using AutoDock Tools.

## 4. Discussion

The anti-hyperuricemic effects of sunflower calathide have recently aroused much attention, and the investigation of its bioactive components against hyperuricemia is of importance. We identified AA from SCE and evaluated its anti-hyperuricemia ability. As a result, AA could inhibit XO in a concentration gradient at low concentrations (IC_50_ = 10.60 µM and Ki = 193.65 nM), indicating a promising inhibitor of XO. Thus, we hypothesized that it would play a crucial role in anti-hyperuricemia therapy and followed a further investigation.

The UA-stimulated HEK293T cell model was established for the investigation of AA (320 µM) and SCE (300 µg/mL) on UA metabolism. The concentrations were determined by our previous research [7] and the results of cytotoxicity. According to the bioinformatic analysis of RNA-seq data, the DEGs of purine metabolism in the AA group were mostly opposite to that in the UA group (Figure 6 and Figure 7). This proves that AA modulates purine metabolism in the UA-stimulated HEK293T cell. However, the SCE was more similar to the control, indicating that AA acts with a different mechanism from SCE.

Purine nucleoside phosphorylase (PNPase) [29], Ribose-phosphate pyrophosphokinase 1 (PRPS1) [30], Ribose-phosphate pyrophosphokinase 2 (PRPS2) [31], and Ribose 5-phosphate isomerase A (RPIA) [32], which play prominent roles in purine biosynthesis and metabolism, were enriched by GO analysis. It is reported that PRPS1 and PRPS2 catalyzed the biosynthesis of phosphoribosyl pyrophosphate (PRPP), which is essential for nucleotide biosynthesis in the pentose phosphate pathway [33]. RPIA is an upstream regulator of them. As a result, the highest abundance of the control group suggested negative feedback regulation after UA stimulation. The AA inhibits purine biosynthesis without cytotoxicity in HEK293T cells, and so does the SCE. Moreover, the AA could bind to PNPase and PRPS1 in molecular docking, suggesting novel targets for hyperuricemia therapy.

This present study revealed AA’s inhibition of key genes of purine/UA biosynthesis and identified new biomarkers for hyperuricemia in human kidney 293T cells. Moreover, AA might contribute to the therapy of any other diseases involving purine biosynthesis and metabolisms, such as obesity and gout. Due to the excellent inhibitory effects of XO, we considered AA as a promising and natural inhibitor for anti-hyperuricemia, but in vivo experiments have yet to be administered. There is no evidence about the effects of AA in vivo, and more animal research that verifies the mechanism needs to be conducted. The gene expression profiles of the hyperuricemic cell model pave the way for further analysis of mRNA changes in human kidneys.

In conclusion, we found that AA is a promising inhibitor for purine or UA biosynthesis. We conclude that AA is one of the active components against hyperuricemia in SCE due to the similar regulation of PNPase, PRPS1/2, and RPIA. This study might provide a new therapeutic strategy for hyperuricemia, and the RNA-seq data of the cell model are a new basis for anti-hyperuricemic drug screening in vitro. The pharmacological activity of AA in other purine disorders ought to be investigated soon.

## Figures and Tables

**Figure 1 molecules-28-05141-f001:**
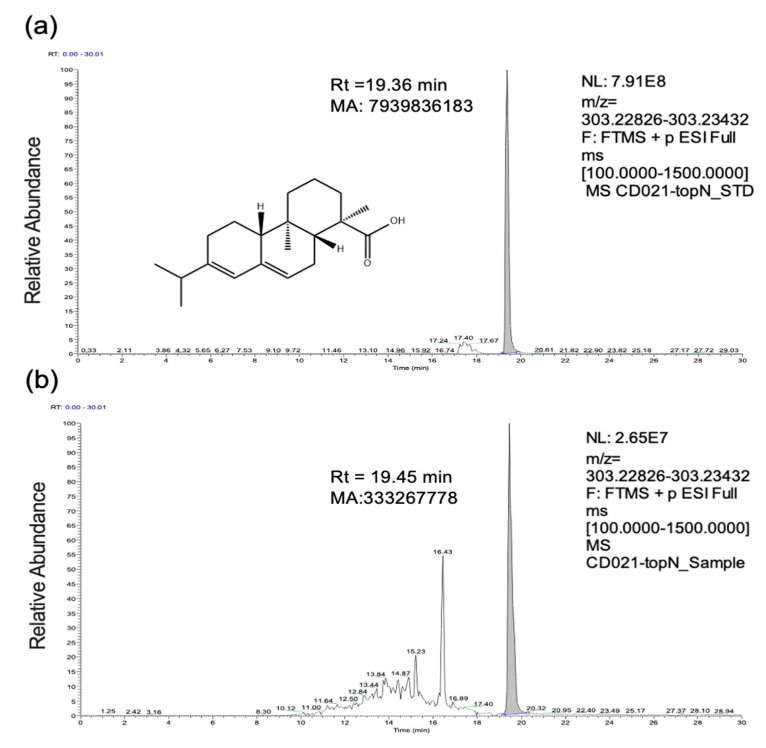
UPLC–MS/MS: chromatograms of standard (**a**) and sample (**b**) with vector plots of relative abundance (measured via Xcalibur 4.1).

**Figure 2 molecules-28-05141-f002:**
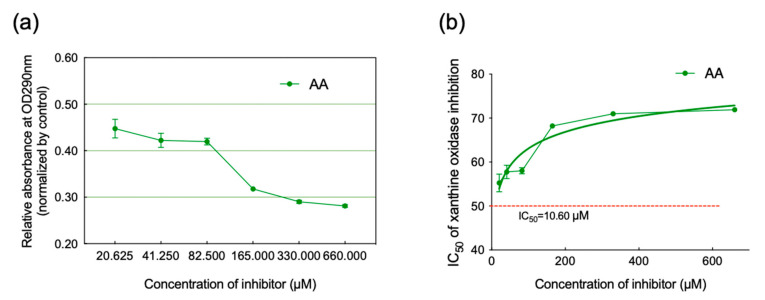
(**a**) Relative abundance of uric acid product (normalized by control: *Y* axis = (A1 − A2)/(A3 − A4)). (**b**) The inhibition curve of AA-XO enzymatic system and the calculation of IC_50_.

**Figure 3 molecules-28-05141-f003:**
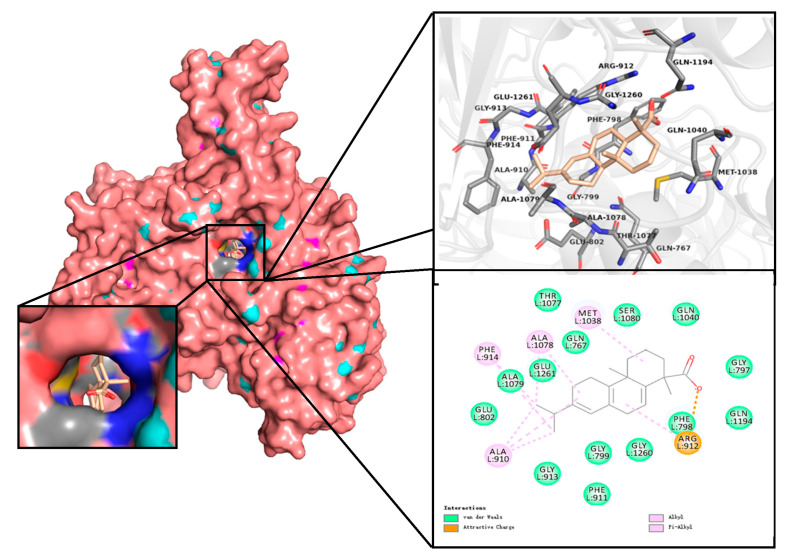
The docked complex with abietic acid and XO. Residues around inhibitor binding pocket. Diagrams were visualized via PyMOL Win and Discovery Studio.

**Figure 4 molecules-28-05141-f004:**
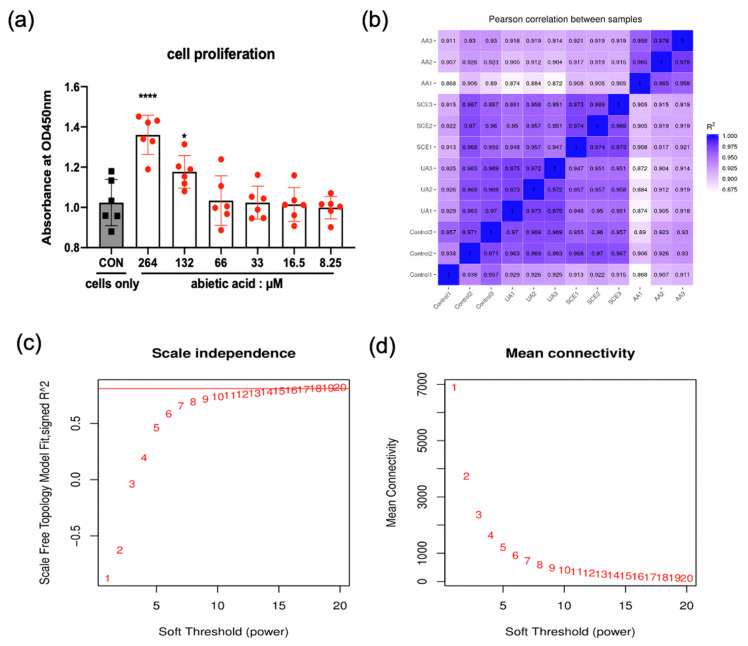
(**a**) Cell proliferation of different concentrations of abietic acid in HEK293T cells. * *p* < 0.05, **** *p* < 0.0001 vs. control group. Each group had no less than six independent experiments. (**b**) Pearson correlation between samples in HEK293T cells. R^2^ represents Pearson’s correlation coefficients (R^2^ > 0.92 indicates reliable biological replicates, as well as R^2^ > 0.8, which suggests intergroup samples are relevant and reasonable based on ENCODE Project Consortium). (**c**) The scale-free topology in the dataset. The red line represents the soft threshold and correlation coefficients (R^2^) corresponding with the WGCNA. (**d**) Mean connectivity of various soft-thresholding powers.

**Figure 5 molecules-28-05141-f005:**
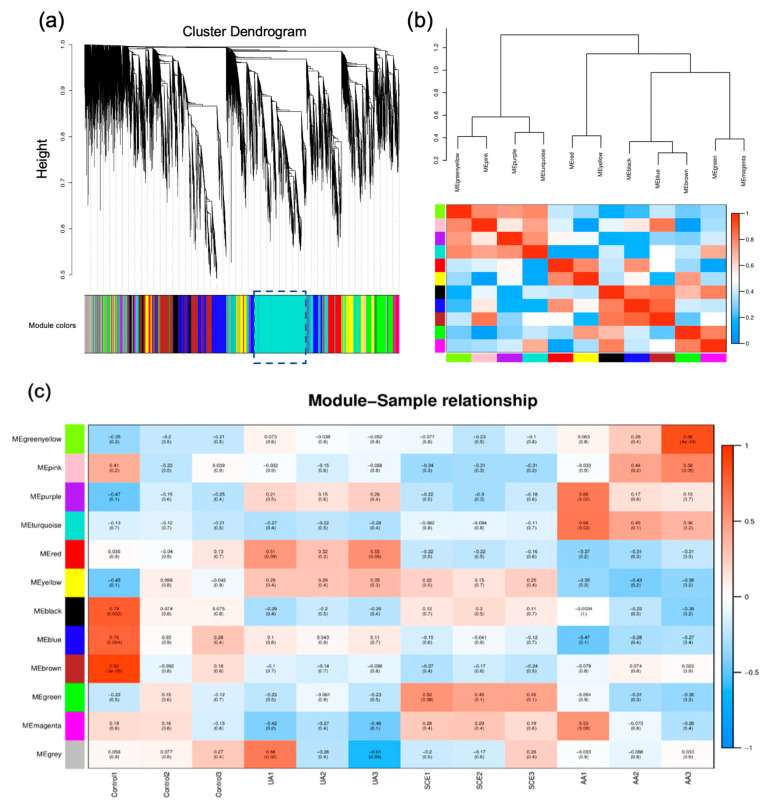
WGCNA of gene expression profiles of different treatments. (**a**) Merged clustering dendrogram of different modules among DEGs. Different colors represent different modules. (**b**) Eigengene dendrogram of module memberships. The ordinate and heatmap indicate correlation coefficients between different modules. (**c**) Relationships between modules and various groups. The related value and *p*-value of correlation were shown in blocks. In the heatmap, red represents a positive correlation, whereas blue represents a negative correlation. The related value and *p*-value were shown in each block.

**Figure 6 molecules-28-05141-f006:**
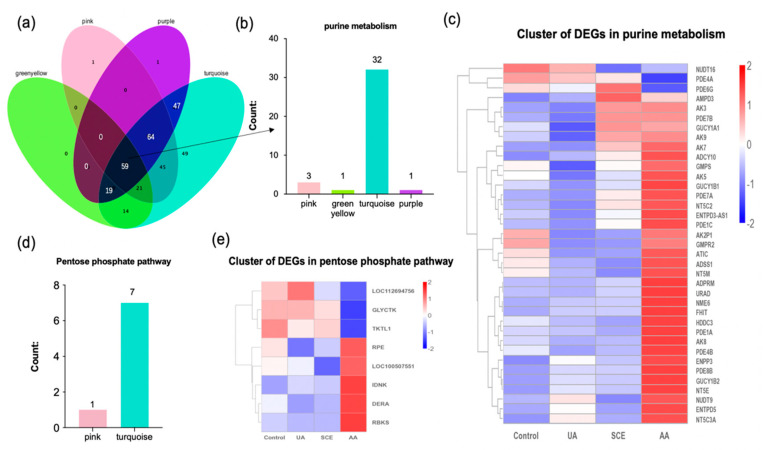
(**a**) Venn diagram among green-yellow, pink, purple, and turquoise modules. Black represented the overlapped KEGG pathways. (**b**,**d**) A statistic of DEGs involving purine metabolism and pentose phosphate pathway. (**c**,**e**) Heatmap of DEGs involving purine metabolism and pentose phosphate pathway among different treatments. Red represents upregulated genes, while blue represents downregulated ones.

**Figure 7 molecules-28-05141-f007:**
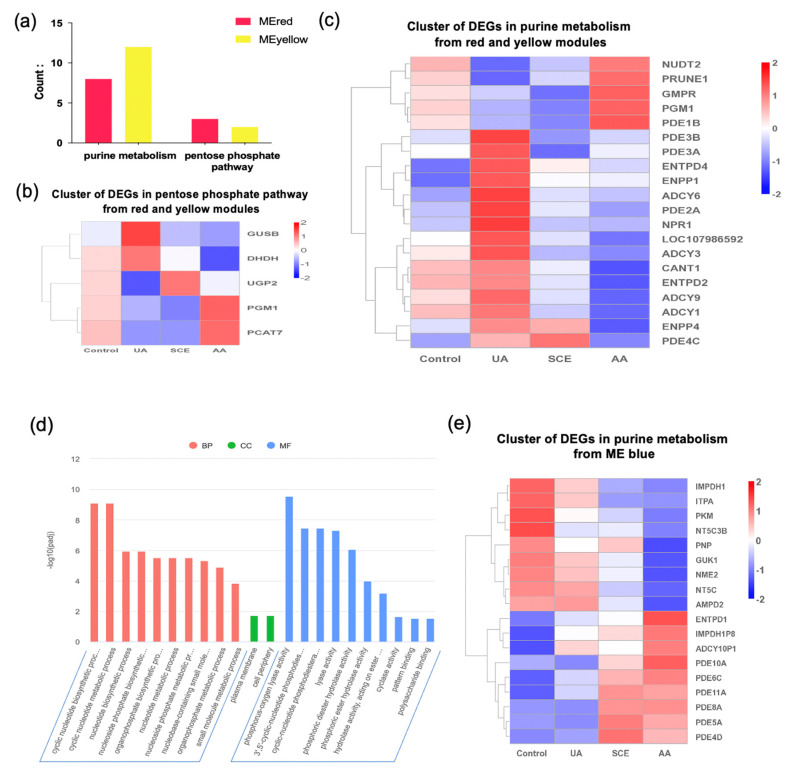
(**a**) Statistical bar graph of DEGs involving purine metabolism and pentose phosphate pathway in red and yellow modules. (**b**,**c**) The heatmap of DEGs in pentose phosphate pathway and purine metabolism among different treatment groups. (**d**) The GO enrichment of downregulated DEGs in AA treatment that shown in Figure 7c. (**e**) The heatmap of purine metabolism-related DEGs in the blue module. Red represents upregulated genes, while blue represents downregulated ones.

**Figure 8 molecules-28-05141-f008:**
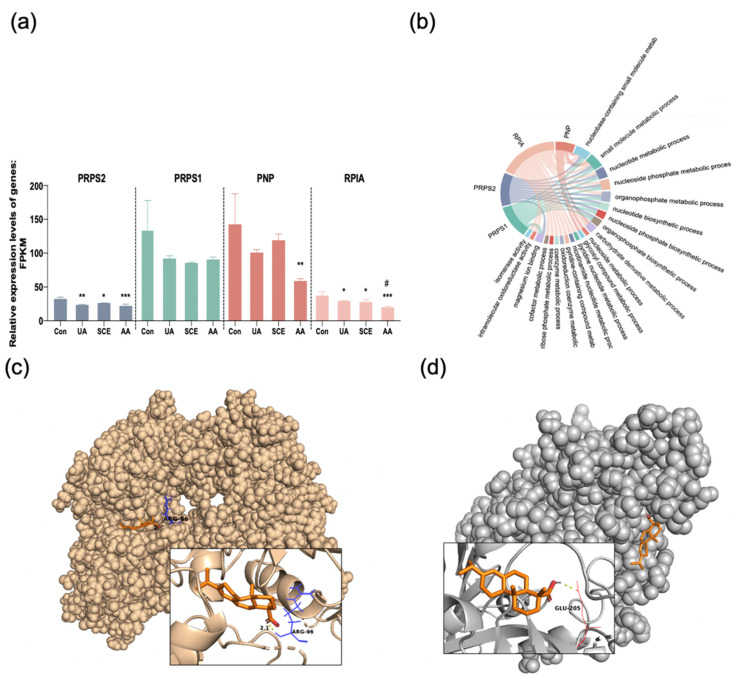
(**a**) The intensity of hub genes in HEK293T cells among different treatment groups. The bar arrow indicates the standard errors. * *p* < 0.05, ** *p* < 0.01, *** *p* < 0.001 vs. control group. *^#^ p* < 0.05 vs. UA group. (**b**) Chordal diagram of four genes along with GO terms. Plotted using R (Version 3.0.3) ggplot2 package. (**c**) Interactions between AA and PRPS1. (**d**) Interactions between AA and PNPase. (Plotted via PyMOL).

**Table 1 molecules-28-05141-t001:** Lamarckian genetic algorithm docking state. (Temperature: 298.15 K).

Sub-Rank	Run	Binding Energy: kcal/mol	Inhibition Constant: Ki, nM	Cluster RMSD
1	3	−9.16	193.65	0.00
2	10	−9.15	195.27	0.03
3	4	−9.15	195.36	0.02
4	6	−9.15	195.48	0.04
5	1	−9.15	197.34	0.03
6	9	−9.15	197.40	0.03
7	5	−9.15	197.94	0.03
8	8	−9.14	198.22	0.03
9	7	−9.13	202.29	0.15
10	2	−9.13	203.26	0.13

## Data Availability

The raw data and data processed in this study are available on the GEO datasets (Accession Number: GSE198133).

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
