# Peer review of "Investigating the Vital Role of the Identified Abietic Acid from Helianthus annuus L. Calathide Extract against Hyperuricemia via Human Embryonic Kidney 293T Cell Model"

_molecules, 2023, doi:10.3390/molecules28135141_

Round 1

Reviewer 1 Report

The paper presents new interesting study results which can be adressed to mamy readers nevertheless some points have to be improved:

1. Abstract should be more informative

2. Methodology should includes more details

3. Discussion is sufficient but its would be valuable if authors include more information about impact of analyzed compounds on human health (more details)

4. Conclusions can be improved 

Author Response

Response to Reviewer 1 Comments

Dear Reviewer 1:

Thank you for your professional comments and we would appreciate your detailed review first. These suggestions help to improve the quality of the manuscript. As you are concerned, there are several problems that need to be addressed. We would like to show the details as follows, and please check the revised version in the attachment.

Point 1: Abstract should be more informative

Response 1: Thank you for your kind comments. We have revised this part according to your advice, please check it in the manuscript.

Point 2: Methodology should includes more details

Response 2: Thank you for your kind comments. We have revised this part according to your advice, please check it in the manuscript.

Point 3: Discussion is sufficient but its would be valuable if authors include more information about impact of analyzed compounds on human health (more details)

Response 3: Thank you for your detailed comments. We have added some detailed information about the impact of abietic acid and please check it in the manuscript.

Point 4: Conclusions can be improved

Response 4: Thank you for your kind comments. We have revised this part according to your comments, please check it in the manuscript.

Reviewer 2 Report

The manuscript presents novel data on the activity of abietic acid. However, before accepting it for publication, I have some suggestions. Overall, I do not understand to why the section on the determination of abietic acid in sunflower calathide was included in the manuscript. It appears that all tests were conducted using pure abietic acid. It should be clarified in the aim of the study.

Other suggestions/questions

-          Title should be rewritten

-          Abstract:  “AA, a natural diterpenoid, is a potential inhibitor” – inhibitor of what?

-          Introduction:

“for example, the abietic acid (AA) [9-11].” – references  are unnecessary here

“Given the chemical structure and abundant bioactivities” - Why did the chemical structure lead to the investigation of this type of activity? More information on bioactivity of AA should be added.

“the regulation of AA” – what did Authors mean?

- 2.1. Preparation of sunflower … - give some detail on extraction

 - 2.2. Characterization by… - characterization of what? Manufacturer of UPLC should be added. Italic for units is unnecessary (correct also in the other sections). A brief description of the chromatographic conditions should be added so that the reader does not have to refer to another publication. Some information on quantification of AA should be added.

-2.3. section: purity of AA is unnecessary. This information is in 2.2 section. ”Spectrophotometer-based determination” – of what? Title of the section should more reflect the content

OD290 nm – what does it mean OD?

-          2.5. section: what solvent was used to dissolve/dilute AA for cell assay? What was the final concentration of solvent? Did it not have an impact on cell viability? The justification for the concentrations  of extract and AA used should be added (e.g. in Discussion). Manufacturer for kit should be added. In what section the results for the experiments are shown? In 3.4. no cytotoxicity results  are shown for UA, SCE groups.

-          Results:  Is the content in μg/mg of dried extract?

-          Table 1 legend: correct the fonts

-          Figure 3: legend is not visible

-          Discussion should be deeper. The results should be discussed with appropriate literature

Language needs correction. Some examples are given below:

“It is well-known for anti-inflammation [12,13] and anti-cancer properties “ – it should be: “for its anti-inflammatory [12,13] and anticancer properties

“Extraction was stored” – extract was stored

“the spectrophotometer method” - the spectrophotometric method

“Dissolved the abietic acid in ethanol and diluted it into six gradients of final concentration” – abietic acid was dissolved in ethanol and was diluted….

“Cell proliferation of different concentrations… “ - Cell proliferation at different concentrations

and many others. Please carefully check the entire manuscript

English needs correction. Some examples are given in Comments and Suggestions for Authors

Author Response

Response to Reviewer 2 Comments

Dear Reviewer 2:

Thank you for your detailed and professional comments. We sincerely agreed with these suggestions and revised them according to your advice. Please check it in the attached manuscript. If there are any other modifications we could make, we would like to correct them and appreciate your kind work.

Reviewer 2:

The manuscript presents novel data on the activity of abietic acid. However, before accepting it for publication, I have some suggestions. Overall, I do not understand to why the section on the determination of abietic acid in sunflower calathide was included in the manuscript. It appears that all tests were conducted using pure abietic acid. It should be clarified in the aim of the study. 

To reviewer 2: Thank you for your detailed concern. We have identified that SCE could lower uric acid levels in mice previously, and in this research, we identified abietic acid in SCE. The aim of comparing abietic acid and SCE is to infer whether they act in a similar mechanism. If they show the same tendency for purine metabolism-related gene expression, we could suggest that abietic acid is the active component of SCE against hyperuricemia. We feel sorry that we may not illustrate this issue clearly. Thus, we would like to discuss much about the aim of comparison in the manuscript. Please check the revised version of the manuscript.

Other suggestions/questions

Point 1: Title should be rewritten

Response 1: Thank you for your kind comment. We have revised the title as " Identified Abietic Acid from Helianthus annuus L. Calathide Extract and Investigate the Vital Role against Hyperuricemia via Human Embryonic Kidney 293T cell model". If you have other advice, we would like to modify it again.

Point 2: Abstract:  “AA, a natural diterpenoid, is a potential inhibitor” – inhibitor of what?

Response 2: Thank you for your kind comment. According to your suggestion, we revised it to " AA, a natural diterpenoid, is a potential inhibitor of purine biosynthesis or metabolism". Please check it.

Point 3: Introduction:

“for example, the abietic acid (AA) [9-11].” – references  are unnecessary here

“Given the chemical structure and abundant bioactivities” - Why did the chemical structure lead to the investigation of this type of activity? More information on bioactivity of AA should be added.

“the regulation of AA” – what did Authors mean?

Response 3: Thank you for your detailed comments. We have revised them as below, please check it. If you have more detailed advice, we would like to make it better.

1) We deleted the reference of [9-11], please check it in the manuscript.

2) We added more references about bioactivities of abietic acid in this part and deleted "the chemical structure", please check it in the manuscript.

3) “the regulation of AA” was modified to "the investigation of AA".

Point 4: 2.1. Preparation of sunflower … - give some detail on extraction

Response 4: Thank you for your kind comment. The added sentence is "The sunflower calathide powder was extracted by cellulase and CaCl2 followed by filtration". In order to avoid showing the same content, we suggested reader refer to our previous paper. If there were any other suggestions, we would like to make it better.

Point 5: 2.2. Characterization by… - characterization of what? Manufacturer of UPLC should be added. Italic for units is unnecessary (correct also in the other sections). A brief description of the chromatographic conditions should be added so that the reader does not have to refer to another publication. Some information on quantification of AA should be added.

Response 5: Thank you for your professional comments. The manufacturer was added and italic units were modified. We have added some information according to the above advice. Please check it in the manuscript and contact us if you have any better suggestions.

Point 6: 2.3.section: purity of AA is unnecessary. This information is in 2.2 section. ”Spectrophotometer-based determination” – of what? Title of the section should more reflect the content.

OD290 nm – what does it mean OD?

Response 6: Thank you for your kind comment. We deleted "HPLC ≥ 98%", and revised ”Spectrophotometer-based determination” to ”Spectrophotometer-based determination of enzyme activity”. The "OD290 nm" means optical density at 290 nm and we have added it to the manuscript. We have revised them in the revised version, please check it.

Point 7: 2.5. section: what solvent was used to dissolve/dilute AA for cell assay? What was the final concentration of solvent? Did it not have an impact on cell viability? The justification for the concentrations  of extract and AA used should be added (e.g. in Discussion). Manufacturer for kit should be added. In what section the results for the experiments are shown? In 3.4. no cytotoxicity results  are shown for UA, SCE groups.

Response 7: Thank you for your kind comment.

1) We revised the above suggestion to the manuscript and illustrated the reason for concentrations in Discussion.

2) Additionally, the reults were shown in 3.4 (Figure 4).

3) According to your comments, we did not show the cytotoxicity results of UA and SCE since that several references had illustrated the cytotoxicity before. We consider it would be repeat content the same as other research.

If you have any other advice, we would like to modify much better.

Point 8: Results:  Is the content in μg/mg of dried extract?

Response 8: Thank you for your kind comment. We feel sorry for the confused characterization. In this part, the μg/mg is the calculation which is converted to sunflower calathide powder rather than dried extract. We have modified it in the manuscript, please check it.

Point 9: Table 1 legend: correct the fonts

Response 9: Thank you for your kind comment. We have corrected it in the manuscript, please check it.

Point 10: Figure 3: legend is not visible

Response 10: Thank you for your kind comment. We feel sorry that the legend was shown on the next page due to typography problems. Here, we displayed it and please check it, or you can review it in the manuscript.

" Fig.3. The docked complex with abietic acid and XO. Residues around inhibitor binding pocket. Diagrams were visualized by PyMOL Win and Discovery Studio."

Point 11: Discussion should be deeper. The results should be discussed with appropriate literature

Language needs correction. Some examples are given below:

“It is well-known for anti-inflammation [12,13] and anti-cancer properties “ – it should be: “for its anti-inflammatory [12,13] and anticancer properties

“Extraction was stored” – extract was stored

“the spectrophotometer method” - the spectrophotometric method 

“Dissolved the abietic acid in ethanol and diluted it into six gradients of final concentration” – abietic acid was dissolved in ethanol and was diluted….

“Cell proliferation of different concentrations… “ - Cell proliferation at different concentrations

and many others. Please carefully check the entire manuscript

Response 11: Thank you for pointing out these mistakes. We have corrected the above mistakes and reviewed the manuscript overall, please check it. Finally, we revised the discussion according to your advice. If you have any other advice, we would like to modify it much better.

Reviewer 3 Report

Minor comment: The authors should include NMR data of abietic acid in the manuscript.

Minor editing of English language required

Author Response

Response to Reviewer 3 Comments

Dear Reviewer 3:

Thank you for your kind and professional advice. These suggestions help to improve the quality of the manuscript. As you are concerned, several problems need to be addressed. We would like to show the details in the revised manuscript and please check it in the attachment.

Comments and Suggestions for Authors

Point 1: Minor comment: The authors should include NMR data of abietic acid in the manuscript.

Response 1: Thank you for your detailed and professional comments. We agree that the NMR would provide additional information about abietic acid. Since the abietic acid used in this study was bought from Solarbio and the content was detected by HPLC over 98%, we did not perform an NMR test. We would like to adopt this suggestion in future research.

Point 2: Minor editing of English language required

Response 2: Thank you for your kind comments. We have rewritten this manuscript according to your advice. Please check it in the attachment.

Round 2

Reviewer 1 Report

Thank you for the paper improvement. The manuscript can be accepted to publication.

Author Response

Dear Reviewer 1:

Thank you for your kind comments. Your professional suggestion made our manuscript much better.

Sincerely yours,

Wannan Li

Reviewer 2 Report

The authors incorporated most of my corrections; however, there are still some points that need to be addressed.

1) It is well-known for its anti-inflammation ….. properties -  should be: antiinflammatory

2) 2.2. Characterization by UPLC-Q-Orbitrap HRMS – should be:  characterization of extract by ….

3) Dissolved the abietic acid (Solarbio, CHN) in ethanol and diluted; Dissolved UA, SCE, and AA in DMEM – The passive form is usually used in publications.: abietic acid was dissolved…. and diluted;   UA, SCE, and AA were dissolved….

4) Figure 3. – figure legend is not visible

Author Response

Dear Reviewer 2:

Thank you for your detailed and professional comments. Your kind work made our manuscript much better. We corrected some mistakes per your suggestion; please check the revised version. 
